# Flaxseed Oil as a Source of Omega n-3 Fatty Acids to Improve Semen Quality from Livestock Animals: A Review

**DOI:** 10.3390/ani11123395

**Published:** 2021-11-28

**Authors:** Jabulani Nkululeko Ngcobo, Fhulufhelo Vincent Ramukhithi, Khathutshelo Agree Nephawe, Takalani Judas Mpofu, Tlou Caswell Chokoe, Tshimangadzo Lucky Nedambale

**Affiliations:** 1Department of Animal Science, Tshwane University of Technology, Private Bag X680, Pretoria 0001, South Africa; NephaweKA@tut.ac.za (K.A.N.); MpofuTJ@tut.ac.za (T.J.M.); NedambaleTL@tut.ac.za (T.L.N.); 2Agricultural Research Council, Germplasm, Conservation, Reproductive Biotechnologies, Private Bag 0062, Pretoria 0001, South Africa; RamukhithiF@arc.agric.za; 3Department of Agriculture, Land Reform and Rural Development, Directorate, Farm Animal Genetic Resource, Private Bag X250, Pretoria 0001, South Africa; TlouC@dalrrd.gov.za

**Keywords:** artificial insemination, antioxidants, fertility, spermatozoa, omega n-3

## Abstract

**Simple Summary:**

In response to the conservation of threatened livestock species, different strategies to improve semen quality have been developed. However, spermatozoa remain sensitive to cryopreservation damages especially that of avian species, thus limiting the use of reproductive biotechnologies such as artificial insemination in the conservation programs. Improving semen quality through dietary inclusion of long-chain polyunsaturated fatty acids sources mainly omega n-3 has received research interest. This review explains the role of flaxseed oil as a source of omega n-3 fatty acids to improve semen quality. Comprehensive information elaborated in this review is believed to promote the use of flaxseed oil as an alternative source of omega n-3 fatty acids to fish oil. This is because fisheries are over-exploited and could collapse.

**Abstract:**

The demand to conserve indigenous species through the cryo-gene bank is increasing. Spermatozoa remain sensitive to cryopreservation damages especially that of avian species thus limiting the use of reproductive biotechnologies such as artificial insemination in the conservation programs. Long-chain polyunsaturated fatty acid (LCPUFAs), specifically omega n-3, expanded a research interest to improve animal reproductive efficiency through improving spermatozoa quality. This is driven by the fact that mammals cannot synthesize omega-3 de-novo because they lack delta-12 and delta-15 desaturase enzymes thus supplemented in the diet is mandatory. Delta-12 and delta-15 add a double bond at the 12th and 15th carbon-carbon bond from the methyl end of fatty acids, lengthening the chain to 22 carbon molecules. Fish oil is a pioneer source of omega n-3 and n-6 fatty acids. However, there is a report that numerous fisheries are over-exploited and could collapse. Furthermore, processing techniques used for processing by-products could complement alterations of the amino acid profile and reduce protein retrieval. Alternatively, flaxseed oil contains ±52–58% of total fatty acids and lignans in the form of α-linolenic and linoleic acid. Alpha-linolenic acid (ALA,18:3n-3) is enzymatically broken-down de-novo by delta-6 desaturase and lengthened into a long-chain carbon molecule such as eicosapentaenoic acid (C20:5n-3). Nevertheless, controversial findings following the enrichment of diet with flaxseed oil have been reported. Therefore, this paper is aimed to postulate the role of flaxseed oil as an alternative source of omega n-3 and n-6 fatty acids to improve semen quality and quantity from livestock animals. These include the interaction between docosahexaenoic acid (DHA) and spermatogenesis, the interaction between docosahexaenoic acid (DHA) and testicular cells, and the effect of flaxseed oil on semen quality. It additionally assesses the antioxidants to balance the level of PUFAs in the semen.

## 1. Introduction

The demand to conserve indigenous species through the cryo-gene bank is increasing [1]. This is driven by the fact that livestock production could be sustained through genetic improvement of vital species together with the control of the diseases affecting their production [2]. Smallholder farmers own sheep, goats, chickens, and cattle among other species to address poverty in developing countries [3]. Through these species, rural farmers produce meat for consumption, maintain household income, produce manure, and graze marginal land not conducive for crop farming [3,4,5], thus playing a huge role in food security [6]. However, there are impediments hampering their reproduction efficiencies. Among various impediments, ±20% of males are incompatible for mating during the breeding season [7]. Moreover, another ±10 to 15% of males show a decreased reproduction efficiency or questionable fertility and semen quality [7].

Artificial insemination (AI) is an alternative tool to improve livestock genetic material and conservation [8]. This, however, requires good quality semen (+70% total motility) before freezing to survive the damaging effects imposed by the cryopreservation processes [9]. Frozen-thawed spermatozoa survival is still challenging in various species [10] making it difficult to perform AI [9,11]. Cryopreservation damages target the sperm plasma membrane making it more permeable. Thereafter, reactive oxygen species (ROS) take advantage and cause harm to the acrosome integrity and sperm motility [12,13]. Sperm cell plasma membrane is made of lipids in the form of polyunsaturated fatty acids (PUFAs) and attached by ROS because of their unconjugated double bonds separated by methylene groups [14]. This occurs due to the lower concentration of sifting enzymes and body defenses confined within the sperm cytoplasm [14]. For that reason, it is pivotal to enhance a balance between lipids, ROS, and antioxidants [15].

Molecular studies have shown the loss of proteins, lipids, and ions during cryopreservation to be a primary cause of poor post-thawed sperm quality [13]. Enriching diets with fatty acids such as omega n-3 has been shown to improve sperm motility [16], sperm membrane fluidity, and reproductive performance [17] in chickens [18], bulls [19,20,21,22], rams [23], and even in buffalos [17]. Ahmad et al. [16], reported that the dietary inclusion of PUFA’s plays a vital role in improving fertilization rate through improving sperm membrane integrity. Khoshniat et al. [22], associated the spermatozoa’s ability to resist cold shock with sperm membrane lipid composition. Hence, there has been a great interest in modifying sperm quality through long-chain polyunsaturated fatty acids (LCPUFA’s) enriched diets indifferent livestock species [10,24,25]. Nevertheless, other authors reported some controversial results following the enrichment of diets with flaxseed oil [22,26]. In their studies, some find no positive effect of feeding flaxseed oil-rich feed on semen quality and fertility. Therefore, this study is aimed to review the role of flaxseed oil as a source of polyunsaturated fatty acids to improve semen quality from livestock animals. Among various livestock animals available, this review is limited to avian, pigs, cattle, buffalo, sheep, and goats.

## 2. Flaxseed Oil Composition

The botanic name for flax is *Linum usitatissimum*, belonging to the *Linaceae* family [27]. It is known that *alsi* or linseed or flaxseed is a good source of α-linolenic and improves the tissue concentration of both α-linolenic acid and eicosapentaenoic acid, which synthesizes pivotal for reproductive hormones [28]. Flaxseed is a source of omega n-3 fatty acids containing about 52–58% of total fatty acids and lignans [29]. Moreover, flaxseed oil contains 58% α-linolenic acid, which is a vital antioxidant source that improves animal health not only reproductively [20,30] but also through improving inflammation and brain development [27]. Flaxseed oil is further comprised of stearic, oleic, linoleic, and palmitic acids, which contain a high content of vitamins including vitamin E [30]. Vitamins such as vitamin E and C (ascorbic acid) play a pivotal role as an antibiotic in the cell.

Flaxseed oil contains precursors (α-linolenic acid) of eicosapentaenoic (EPA) and docosahexaenoic (DHA), which are broken down de novo to form long-chain PUFAs [30] and later converted to docosahexaenoic acid which is vital for testes’ functioning [18,31] (Figure 1). In human beings, flaxseed oil protects the body against cardiovascular disease, inhibits pro-inflammatory mediators, reduces low-density lipoprotein (LDL) cholesterol, plays a significant role in bone health, and reduces hormone-related cancers [32]. Furthermore, flaxseed oil increases the level of α-linolenic and eicosapentaenoic acid that are important in the synthesis of reproductive hormones [27]. Essential α-linolenic acid (ALA, 18:3n-3) is enzymatically broken down de novo by delta-6-desaturase and lengthened into long-chains such as eicosapentaenoic acid (C20:5n-3) [30], stearidonic acid (SDA) eicosapentaenoic acid (EPA, 20:5n-3), and converted into docosapentaenoic acid then to docosahexaenoic acid (DHA, 22:6n-3) [31]. In the n-6 family, essential linolenic acid (LA, 18:2n-6) is altered into long-chain arachidonic acid (AA, 20:4n-6) [31].

## 3. Classifications and General Functions of Long-Chain Polyunsaturated Fatty Acids

Long-chain fatty acids can be distinguished into three groups: n-3, n-6, and n-9, depending on the location of the first double bond [30]. Most n-9 fatty acids (FA) are mono-unsaturated fatty acids including oleic acid. Polyunsaturated fatty acids involve n-3 and n-6, which are distinguished with more than one double bond between their carbon atoms [35,36]. In cooperation, n-3 and n-6 are classified as essential fatty acids because mammals cannot synthesize them de-novo.

Generally, PUFA’s have several benefits in the body and in the reproductive cells. These benefits range from maintaining the structure and functions of the cell membrane, enhancing immune function, promoting growth and development, regulating lipid metabolism and related gene expression, and in the reduction of thrombosis [17]. Omega-3 fatty acids play a significant role in human health through immune system responses [37,38,39], controlling chronic diseases such as cardiovascular diseases [40] and cancer [41], act as bioactive cellular components of membrane phospholipids [35,40,42]. Therefore, supplementing PUFAs to animals’ diet would also be beneficial to human health because omega-3 FA fed to the animal might later be found in their meat or milk. 

PUFAs can be retrieved from all-natural foods but vary in quantities, with fatty foods and some seafood containing extra. Vegetable oils such as sunflower oil, safflower oil, corn oil, flax oil, sesame seed oil, pumpkin seed oil, and rapeseed oil are also sources of PUFAs. Moreover, they are also found in marine fish such as mackerel, salmon, sardine, herring, and smelt. However, there are alarming studies in some countries like Indonesia where serious fish over-exploitation is reported [43]. Furthermore, processing techniques used for processing by-products could complement alterations of the amino acid profile reduce protein recovery [44]. Therefore, diverting to plant-based sources will be advantageous should a similar quality be maintained.

## 4. Interaction between Docosahexaenoic Acid (DHA) and Spermatogenesis

Docosahexaenoic acid (DHA, C22: 6n-3) is a long-chain omega n-3 and widespread unsaturated fatty acid found in mammalian cells [45]. It is synthesized from its precursor’s alpha-linolenic and linoleic acid with the aid of delta-5 and delta-6-desaturase enzymes [46]. Docosahexaenoic acid accounts for 10% of the brain phospholipids and has a high content in the central nervous system, distributed in acetylcholine, amino phospholipids, and serine glycerol [47]. If DHA concentration in the spermatozoa is lower than that of normal level is associated with infertility [30,48,49]. Docosahexaenoic acids are major elements for human and ruminants’ spermatozoa phospholipids [50]. DHA makes up 30% of esterified fatty acids in phospholipids and 73% of all PUFAs [7]. Therefore, neurons, photoreceptors, and spermatozoa are the cells richest in DHA content [47].

Docosahexaenoic acid has numerous functions such as the role in brain cell function to improve communication between the brain cells. Thus, the lack of n-3 PUFAs in the body can cause a brain communication breakdown [51]. It also appears that DHA influences the membrane structure of the sperm cells by providing essential sperm membrane fluidity and participates in cell response mediation in protein [47]. The DHA concentration in the testicles supports the role of astrocytes, retinal pigment epithelial cells, and Sertoli cells [52]. This could in turn affect the production of lipid-mediated conductors, cell signal transduction, and gene expression. Furthermore, DHA is known to be immunomodulatory, which can affect both innate and adaptive immunity and be used in developing treatments for inflammatory diseases such as rheumatoid arthritis, psoriasis, and ulcerative colitis [41]. Testicular cells especially the Sertoli cells and spermatozoa, are rich in polyunsaturated fatty acids and a fatty acid-enriched diet can control fatty acids profiles in the reproductive tissue [50]. Spermatogenesis involves a sequence of proliferative phases, differentiation, cell divisions to mitotic, meiotic, and spermatogenic stages [53].

Different stages of spermatogenesis include spermatogonia, spermatocytes, and spermatids. Sertoli cells regulate spermatogenesis through playing a vital role in the endocrine and paracrine and providing seminiferous tubules (that manufacture spermatozoon) with nutrition, transporting mature spermatids to the seminiferous tubules’ lumen, discharging androgen-binding protein, and cooperate with Leydig cells for sperm production [53]. During these spermatogenesis stages, lipid droplets occur throughout the process [54], suggesting a very close link between lipid metabolism and fertility during spermatogenesis [53].

Docosahexaenoic acid influences spermatogenesis and sperm quality (Table 1). It is believed that 99% of DHA is found in the tail of the sperm cell [47]. This suggests that DHA in the sperm tail is associated with fluidity and flexibility as well as movements or sperm motility. However, other authors of studies in bulls and humans suggested that a high concentration of DHA is rather found in the sperm head than the tail [55,56]. This suggests the role of DHA in nuclear transfer and fusion with zona-pellucida.

## 5. Interaction between Docosahexaenoic Acid (DHA) and Testicular Cells

Testicular cells, particularly Sertoli cells, are active in the conversion of 18 and 20 carbon omega n-3 into 22 carbon omega n-3 PUFAs [68] because testes have high levels of desaturase mRNA and elongate enzyme. However, higher PUFAs have been reported in germ cells [69]. Germ cells are higher in PUFAs than Sertoli cells but Sertoli cells can convert essential fatty acids (LA and ALA) to derivatives DPA and DHA more than germ cells, hence there are higher 5 and 6 desaturase enzymes in Sertoli cells than germ cells [52]. Testicles have high capacities for the conversion of C20:5n-3 to long C22:6n-3 due to an active elongation and desaturase enzymes in the Sertoli cells [68]. These make testicles an extraordinary organ for PUFA (LA and ALA to derivatives ARA, EPA, DPA, and DHA) metabolism like that of the liver organ [52]. However, when spermatozoa transit to the epididymis for storage, caudal PUFAs are continuously drained from the testicles. Moreover, unsaturated fatty acids sometimes are altered by hormones such as luteinizing hormones (LH) and or adrenocorticotropic hormone (ACTH) in the testis by altering the functions of the enzymes [52].

## 6. Comparison between the Flaxseed and the Fish Oil to Improve Fresh Semen Quality of Livestock Animals

Fish oil is a common source of omega n-3, specifically DHA and EPA [50,68]. Omega n-3 sourced from fish oil has been found to improve semen from rams [10,69], roosters [70,71], and bucks [72,73] through improving total motility [72,74,75,76]. Nevertheless, numerous problems associated with feeding fish by-products have been articulated [77], leading to an interest in replacing fish by-products with plant-based by-products [78]. Therefore, plant base by-products such as flaxseed oil is essential as an alternative to fish by-products in providing necessary omega n-3 fatty acid.

The effects of flaxseed oil on semen quality have been investigated (Table 1) in rabbits [33], avian [18,58], and cattle [22,79,80]. Furthermore, flaxseed oil modulated semen production, quality, quantity, freezability, testicular biometrics, and endocrinological profiles in Mithun bulls [20]. However, the suitable quantity of flaxseed oil necessary for each species is still controversial [18,58,79,80,81]. Noteworthy, depending on the species, sperm motility decreased with the increase of flaxseed oil percentage in the diet (Table 2).

## 7. Flaxseed Oil Alteration between Monogastric and Ruminant Livestock

Ruminants have low fatty acids intake ranging between 2.5–3.5% [Table 2]. Based on the information above (Table 2), avian needs only 2% of flaxseed oil which is the same as that of fish oil to improve semen quality. This was evident when 2 and 4% inclusion of flaxseed was used and was concluded that semen was improved in the 2% group than in the 4% treatment group [71]. This supports a hypothesis that high flaxseed oil is detrimental or has no effect on semen quality especially when no natural antioxidants are used to cause an imbalance, suggesting a necessity to add vitamin E or C on the diet [10]. In pigs, low (3%) flaxseed oil is necessary to improve semen quality and fertility [81]. However, there is still a lack of information with regards to pigs when flaxseed oil is reduced to below or higher than 3% and when no antioxidants are used. In large ruminants (cattle, buffalo), flaxseed oil seems to bypass rumen when dosed [83]. 

## 8. Role of Dietary Inclusion of Omega n-3 on the Post-Thawed Sperm Quality

Cryopreserving semen is essential, particularly because it facilitates the global trade of semen and enables the long-term storage of sperm from superior sires [84,85]. Furthermore, it subsidizes the enlargement of assisted reproductive techniques such as AI and in-vitro fertilization [86] and is useful in difficulties such as extinction, infertility, and injury [87]. However, sperm cells succumb easily to cryopreservation, with subsequent irreversible motility loss [88]. Irreversible motility loss caused by the lipid peroxidation and deoxyribonucleic acid (DNA) damage leading to the less fertilizing ability of the sperm cells [87].

Benefits to supplement dietary fish oil on semen quality of livestock animals through improving sperm plasma membrane of post cryopreservation has been reported [10,80,88]. There was improvement on post-thawed sperm motility in chicken [89], bulls [90], and bucks [87] following supplementation with flaxseed oil. Nevertheless, flaxseed oil-treated bulls produced high total and progressive motility in frozen-thawed semen despite the high content of DHA in fish oil [30]. Moreover, flaxseed supplementation led to 47.8% total post-thawed goat sperm motility [66]. 

## 9. Role of Antioxidant against PUFAs’ Vulnerability to Oxidation

Testicles contain reproductive tissues and cells that are involved in spermatogenesis with a clear role in long-chain polyunsaturated fatty acids (LCPUFAs) production [91]. Nevertheless, testicles are considered oxidation-sensitive organs due to higher PUFAs [92]. It is well known that PUFAs are susceptible to oxidation via free radicals [92] due to their double bonds between the carboxyl chain [93]. Although seminal plasma contains reactive oxygen species (ROS) scavengers, dietary inclusion of PUFAs cause an imbalance of the antioxidant. Therefore, the inclusion of antioxidants is necessary [94]. See Figure 2 which illustrates the process of PUFA supplementation. There is still not enough information on the effect of antioxidants to defend spermatozoa from oxidative stress when flaxseed oil is used as a source of PUFAs. However, on the fresh, the effectiveness of antioxidants seems to be species-specific when supplemented with flaxseed oil (Table 3). In short, when flaxseed oil is increased with no antioxidants, semen quality decreases in avian [18]. Nevertheless, no effect of increasing flaxseed oil concentration in Buffalo bulls [79] sheep [84], and cattle [78] even without an antioxidant. As previously mentioned, flaxseed has some antioxidants in itself scavenging ROS [30]. Nevertheless, more studies evaluating the role of antioxidants in sperm quality when supplemented with flaxseed oil in livestock are still needed.

Antioxidants are in the form of enzymes that absorb hypothetically lethal ROS such as hydrogen peroxide (H_2_O_2_), superoxide anion, and small molecular mass scavengers that can terminate free-radical-mediated chain reactions [95]. The enzyme antioxidants involved in H_2_O_2_ metabolism are the glutathione peroxidase system and/or catalase. Those involved in superoxide anion include superoxide dismutase and/or sometimes indoleamine dioxygenase. Moreover, vitamin C, E, and a variety of polyphenols are classified as small molecular mass scavengers that aid in terminating free radical-mediated chain reactions [95]. Their function is to protect ejaculated spermatozoa from the notorious effects of ROS [96].

## 10. Conclusions

Long-chain polyunsaturated fatty acids (LCPUFAs) are of current prodigious interest in livestock species to improve semen quality. This review described that dietary inclusion of flaxseed oil improves semen quality in livestock animals. Flaxseed oil provides adequate alpha-linolenic as a precursor to omega n-3 through desaturase enzymes, to form long-chain polyunsaturated fatty acids, and hence can be a good alternative to fish oil. This may further address the issues related to fish sustainability and the contamination with meat and bone products during the manufacturing processes. Nevertheless, most of the data has been tested using in-vitro spermatozoa quality, hence more work is needed to test in-vivo fertility e.g., lambing, calving, and the farrowing rate following feeding with flaxseed oil. Conversely, the flaxseed oil quantity necessary for superior results differs in each species. Therefore, it is suggested that more studies are conducted to evaluate the different concentrations of flaxseed oil and the role of antioxidants in each species. This will make a comprehensive conclusion on whether LCPUFAs improve livestock production or not which may be helpful particularly for conservation purposes and disseminating semen from superior sires globally through cryopreservation and artificial insemination.

## Figures and Tables

**Figure 1 animals-11-03395-f001:**
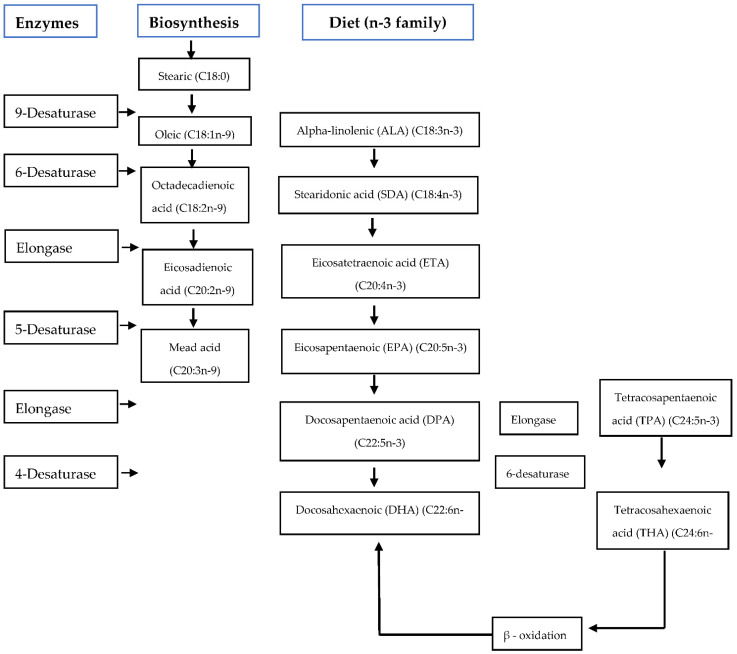
De-novo biosynthesis and the transformation of the omega n-3 family PUFA’s by desaturation and elongation [33,34].

**Figure 2 animals-11-03395-f002:**
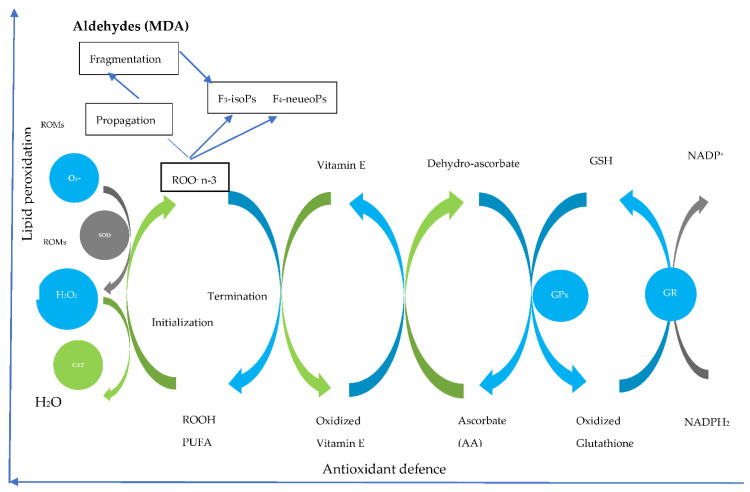
Steps of the omega n-3 oxidative chain and antioxidant interactions [92]. MDA-malondialdehyde, ROMs-reactive oxygen species, GSH-glutathione, GPX-glutathione peroxidase, GR-glutathione reductase, F3isoPs-Isoprostanes, F4-NeuroPs-F4-Neuroprostanes, SOD-superoxide dismutase, CAT-catalase.

**Table 1 animals-11-03395-t001:** Influence of dietary supplementation with PUFAs sourced from flaxseed oil in livestock animals’ reproduction.

Species	Effect	Reference
Avian	Improved semen quality and fertility in broilers	[18,57,58,59]
Pigs	Increase sperm concentration, antioxidant capacity and sperm quality and fertility	[49,60,61]
Cattle	Improved testicle development, spermatogenesis, sperm motility and viability, improved post thawed sperm quality	[27,62]
	Improved fertility in heat stressed Holstein breed	[63]
	Improved post-thawed sperm quality (motility, progression and velocities)	[64]
Buffalo	Improved testosterone concentration and reduced age at puberty	[17]
Goats	Improved sperm motility, vitality, number of sperms with intact plasma membrane after frozen and thawed.	[65]
	Improved frozen-thawed semen quality	[66]
Sheep	Improved sperm quality and quantity and extended semen quality after the breeding season	[67]

**Table 2 animals-11-03395-t002:** Comparison between the flaxseed and the fish oil to improve fresh semen quality of livestock animals.

Species	Oil Source	Supplementation	Effects	Reference
Avian	Flaxseed	2%	Improved sperm concentration, motility and membrane integrity	[18]
	Flaxseed	2%	Improved reproductive hormones in aged rooster.	[71]
	Flaxseed	4%	No effect.	[71]
	Fish	2%	Improved cold stored sperm motility at 24 h	[70]
	Fish	15 g/kg	Improved post thawed semen through reducing apoptosis	[80]
Pigs	Flaxseed	3%	Improved seminal plasma composition, semen quality and farrowing rate	[81]
Cattle	Flaxseed	2% and 4%	Improved testosterone and semen quality	[78]
Buffalo	Flaxseed	125 mL and 250 mL/day	250 mL improved semen parameters better than 125 mL	[79]
Goats	Flaxseed	30 g/kg [3%]	Improved frozen thawed sperm motility	[66]
	Fish	2.50%	Improved semen quality and fertility	[33]
Sheep	Flaxseed	8%	Stimulated seminiferous tubules development and improved the number of Sertoli cells	[82]
	Flaxseed	10%	Reduced age at puberty, improved sperm motility and concentration	[75]
	Fish	2.50%	Improved only semen volume	[69]
	Fish	3%	Did not improve frozen-thawed sperm quality	[10]
	Fish	3%	Reduced negative effect of season on the sperm quality	[68]

**Table 3 animals-11-03395-t003:** The role of antioxidant against PUFAs’ vulnerability to oxidation.

Species	Flaxseed Oil	Antioxidant	Effects on the Semen Quality	Reference
Avian	2%	Vitamin E	Improved sperm concentration, motility and membrane integrity	[18]
	2% and 4%	No antioxidant	2% flaxseed oil improved reproductive hormones in old rooster. Whereas 4% flaxseed oil could not perform better than 2%.	[71]
Pigs	3%	No antioxidant	Improved seminal plasma composition, semen quality and farrowing rate	[81]
Cattle	2% and 4%	No antioxidant	Improved testosterone and semen quality	[78]
Buffalo	125 mL and 250 mL/day	No antioxidant	Semen parameters were improved with the increase of flaxseed oil dosage	[79]
Goats	3%	Vitamin E	Improved frozen thawed sperm motility	[66]
Sheep	8%	No antioxidant	Stimulated seminiferous tubules development and improved the number of Sertoli cells	[72]
	10%	No antioxidant	Reduced age at puberty, improved sperm motility and concentration	[84]

## Data Availability

No new data were collected or analyzed in this study.

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
