# Peer review of "Flaxseed Oil as a Source of Omega n-3 Fatty Acids to Improve Semen Quality from Livestock Animals: A Review"

_animals, 2021, doi:10.3390/ani11123395_

Round 1
Reviewer 1 Report
In the paper “Flaxseed oil as a source of omega n-3 (docosahexaenoic acid, 2 DHA) fatty acids to improve semen quality from livestock: A review” the authors presented information concerning effect of omega n-3/DHA on semen quality parameters of different farm animals.
According to the title authors should focus on flaxseed oil, its composition, and its effect on the semen quality. Then the effect of flexeed oil/ omegan-3/DHA should be described in regards to reproductive performance and sperm quality of the farm animals. All information not connected with reproduction and farm animals should be removed.
The authors should also describe the effect of dietary flaxseed oil on semen quality after cryopreservation as suggested in the abstract. Finally, some mechanism of action of flexeed oil/omegan-3/DHA on spermatozoa should be suggested by the authors.
Author Response
General message from the Authors to the Reviewer.
Dear Reviewer,
We thank you for reviewing our work and your constructive comments that you gave us. It really assisted us to can improve our work. Kindly find the document enclosed herewith for your perusal. We hope that all comments were addressed.
Regards
Manuscript Authors

Reviewer 2 Report
Manuscript needs correction. In the „Simple summary” there is reference to marine fish oil and in the manuscript there is not enough of comparision with linseed oil.
The title of the manuscript would be more appropriate if it was without DHA… flaxseed oil is the source of omega n-3 and that statement should be enough… The transformations of PUFA have been explained in the text.
In the opinion of the reviewer, the subject of the work was not fully covered. The layout and content of the work require some thought. There are repetitions in the text (e.g. L211-214 and L225-227). There is a lack of clarity and some paragraphs are very similar to each other. The sentence L115-117 requires a short extension to include information about the LCFA transformations in monogastric and polygastric animals. Greater accuracy is also expected in the case of uniformity of notation in the text, e.g. delta - 6 desaturase and Δ-6 desaturase and some more.
Table 1.
The name of the column should be corrected (spelling mistakes) The 2. Column is not neccessary it all the shown studies are relevant to C22:6 like in the title of the table. There should be given the correct names of the species – cattle instead of bulls, sheep instead of ram or authors should change the name of the column. Maybe it will be better to add in the table the information about the source of DHA.
Table 2.
If th title of the table is correct if the Authors give an information about growth performance and immune reponces…
What is the difference betwen the table 1 and 2?
The name of the columns in the table 2 needs correction: column 1 – application? Or species?
Type of PUFA’s or source?
It will be good to add maybe some information about the fish oil.
L179 …. Testicales have high capacities for the desaturation and elongation….” It is too much of a mental shortcut.
L188 sentence „Male feritility is crucial because the ratio of males to females is very low” will be good to add some information what does it mean from animal genetic resources conservation point of view.
L225-227 sentence „ Vitamins such as vitamin E and C play a pivotal role as an antibiotic in the cel – requires an explanation or the source of the literature
Maybe it will be good idea to give the diagram of PUFA changes and to check the information about the tranformation of PUFA because it was repeated several times.
Conclusion
Conclusion need to be corrected.
Its execution requires a sentence „This may address the issue of cannibalism when another animal based oil such as marine fish oil is used” What does this have to do with the preceding paragraph?
Is it really true that there are not information about the effect of linseed oil on reproduction parameters? There is a lot of publication on that topic and there is a lot evidence that linseed improve the livestock production so maybe Authors have here something different on their mind or should be more precise defining the aims of next studies.
The list of publications used to prepare the work includes 109 items, of which over 40% are from the last 5 years. The list of publications needs to be corrected. There are spelling mistakes, in some cases, the publication year is missing (e.g. item 46; 54; 81,82, 92, 99), or the journal records are not uniform. Some items are unnecessary as they are only used for general statements and only once. The manuscript relates to livestock and the list of references contains references to publications on elephants (item 3), mice and rats (item 64; 65, 105, 107) or human. In the review it will be good to avoid using the items such as encyclopedias and other review papers.
Taking the mentions above the paper is suitable for publication in the “Animals” after the major revision.
Author Response

(The authors gave the same response as above.)

Round 2
Reviewer 1 Report
This is for a revised manuscript by Ngcobo et al. Overall quality was improved. However, there are still several insufficiencies that need to be corrected.
According to my opinion the Flaxseed oil composition should be placed at the beginning of manuscript as the second section. This part introduce to Docosahexaenoic acid section.
Please justify the presence of Overview of the fatty acids (FA) section. The knowledge regarding FA should be presented in reference to flaxseed oil.
I cannot see any connection with Flaxseed oil in the following sections: 4. Influence of dietary supplementation with PUFAs in animal reproduction; and 8. Antioxidants to balance PUFAs level in semen.
In the section 6. Comparison between the fish oil and the flaxseed oil to improve fresh semen quality of livestock animals, some mechanism of action of flexeed oil/omegan-3/DHA on spermatozoa should be be suggested.
All used abbreviation should be explained when used for the first time.
Table 1 need to be correctly edited.
Author Response
The general message from the Authors to the Reviewer.
Dear Reviewer,
We thank you again for reviewing our work and the constructive comments that you gave us. It really assisted us to can improve our work. Kindly find the document enclosed herewith for your perusal. We hope that all comments were addressed.
Regards
Manuscript Authors

Reviewer 2 Report
Manuscript still needs correction.
In Simle Summary – is AI an advanced reproductive biotechnologies?
L27 –Greater accuracy is also expected in the case of uniformity of notation in the text, e.g. „delta12 and delta-15 desaturase enzymes” – needs unification in the form; the same with monounsaturated and mono-unsaturated, polyunsaturated and poly-unsaturated….
L55 „twenty percent” and ± 10 to 15% as above
Table 1 should be prepared more aesthetically, the same figure 1
In the Figure 1 spelling mistake with C18:3, n-3
L300 Vitamin C instead of vitamin C, and vitamin c in 8.1
In some part of manuscript there are repeated information about the reducing effect of linseed, omega 3 etc on cariovascular diseases and cancer.
There is still lack of clarity and some paragraphs are very similar to each other. In reviewer opinion the Authors should start from some general information – about flaxseed and afterthat they suppose to give paragraphs about the fatty acids and its role in reproduction, testical cells, spermatogenesis, post-thawed sperm quality
If the article is about the lifestock there should be information about changes between polygastric and monogastric animals because the effect of linseed will be less „spectacular” in the case of ruminants… and maybe it will be connected with the doze of the n-3 FA source in the diet.
Is it really true that the linoleic acid C18:2, n-6 is the precursor of DHA? Point 3
Taking the mentions above the paper is suitable for publication in the “Animals” after the minor revision.
Author Response

(The authors gave the same response as above.)
